## COMMENT

# Ethnic disparities in publicly-available pulse oximetry databases

Fatemeh Y. Sinaki [1], Rabab Ward [2], Derek Abbott [3], John Allen [4], Richard Ribon Fletcher[5,6], Carlo Menon [7,8] & Mohamed Elgendi [1,2,7,8]✉

Inaccuracies have been reported in pulse oximetry measurements taken from people who identified as Black. Here, we identify substantial ethnic disparities in the population numbers within 12 pulse oximetry databases, which may affect the testing of new oximetry devices and impact patient outcomes.

There has been an unprecedented demand for pulse oximetry—a method of determining the oxygen saturation ($SpO_2$) of the blood—during the COVID-19 pandemic to aid in medical decision-making. Pulse oximetry data are also widely used for medical research and algorithm development. The measurement of SpO2 involves shining light onto tissue at two separate wavelengths and derives the oxygen saturation from the relative changes in light absorption with each heartbeat. This pulsatile component is independent of skin pigmentation; however, other factors, such as the specific properties of the light source and the algorithms used by the product manufacturer, can produce variations that depend on skin pigmentation[1].

A recent study revealed the clinical importance of racial disparities in pulse oximetry readings[2]. Specifically, when compared to measurements of arterial oxygen saturation, the pulse oximetry algorithms in these devices were found to produce systematically higher saturation values in Black patients compared to white patients. Such systematic racial biases could adversely affect clinical decision-making, such as triage for supplemental oxygen, due to the pulse oximetry readings of Black patients appearing artificially higher. Such inaccuracies may disproportionately increase the risk of unrecognized low oxygen saturation levels in Black patients under certain circumstances, for example in people with COVID-19[3].

Recent advancements in artificial intelligence (AI) have relied on using public databases to undertake feature extraction with pulse oximetry signals to assess hypertension[4], estimate lung function[5], and validate algorithms developed for monitoring patients with COVID-19[6]. To prevent potential disparities in the calibration and accuracy of pulse oximetry devices and their algorithms, the pulse oximeter signals within such public databases need to be representative of the diverse populations on which these devices are used.

Since inaccuracies in pulse oximetry readings have been attributed to differences in skin pigmentation and skin pigmentation varies with race and ethnicity, it is essential to clarify these terms as used in this article. We have predominantly chosen to use the term ethnicity because that is the specific term used in the public datasets and is also the term that is identified by the patients themselves. For the purpose of discussing health disparities, while both race and ethnicity are social constructs, ethnicity has emerged as the preferred one since it encompasses cultural aspects of social identity[7] that extend beyond the more simplistic view of race that is primarily based on shared skin pigmentation or physical characteristics[8]. While members of a

[1] Rady Faculty of Health Sciences, University of Manitoba, Winnipeg, Manitoba R2H 2A6, Canada. [2] School of Electrical and Computer Engineering, University of British Columbia, Vancouver, BC V6T 1Z4, Canada. [3] School of Electrical and Electronic Engineering, The University of Adelaide, Adelaide, SA, Australia. [4] Research Centre for Intelligent Healthcare, Coventry University, Coventry CV1 5RW, UK. [5] Mechanical Engineering, Massachusetts Institute of Technology, Cambridge, MA, USA. [6] Department of Psychiatry, University of Massachusetts Medical School, Worcester, MA, USA. [7] School of Mechatronic Systems Engineering, Simon Fraser University, Burnaby, BC V5A 1S6, Canada. [8] Biomedical and Mobile Health Technology Laboratory, Department of Health Sciences and Technology, ETH Zurich, 8008 Zurich, Switzerland. ✉email: moe.elgendi@hest.ethz.ch

given ethnicity can express a range of skin pigmentation, it is generally agreed that those patients who self-identify as Black generally have a darker skin pigmentation than other ethnic groups.

In order to investigate the proportion of individual ethnicities represented in publicly available pulse oximetry databases, we conducted a comprehensive assessment of accessible databases from 1st January 2012–1st January 2022 using PubMed consisting of Medical Subject Headings (MeSH) terms and Title/Abstract keywords. Applying the inclusion and exclusion criteria defined in Fig. 1 resulted in 12 research articles describing 12 publicly-available datasets to assess different medical conditions using pulse oximeter data.

In total, as of January 28th 2022, these databases have been used to produce over 6214 citations according to Google Scholar including 3544 citations for Medical Information Mart for Intensive Care (MIMIC III)[9]; 1049 citations for MIMIC II[10]; 531 citations for IEEEPPG Dataset[11]; 243 citations for Multiparameter Intelligent Monitoring in Intensive Care I (MIMIC I)[12]; 239 citations for WESAD[13]; 215 citations for Vortal Dataset[14]; 102 citations for the CapnoBase Dataset[15]; 87 citations for the University of Queensland Vital Signs Dataset[16]; 86 citations for PPG-DaLiA[17]; 63 citations for PPG-BP Dataset[18]; 50 citations for Wrist PPG Signals Recorded during Exercise[19]; and 5 citations for Medical Information Mart for Intensive Care IV (MIMIC-IV)[20]. We evaluated the existence of potentialdisparities in ethnicity based on the existing patient records as reported in the publicly available databases. In the absence of such information, the numbers of subjects of each category were inferred and quantified based on the authors' research institutions' locations or where the data was collected, as shown in Table 1.

To avoid any uncertainty in the results of ethnic disparity analysis for a given population, databases with inferred ethnicity information were excluded from the statistical analysis. Four databases for which data for ethnicity was clearly stated, MIMIC, MIMIC-II, MIMIC-III and MIMIC-IV, were included in the statistical analysis. The distribution of ethnic groups in the four databases is shown in Fig. 2.

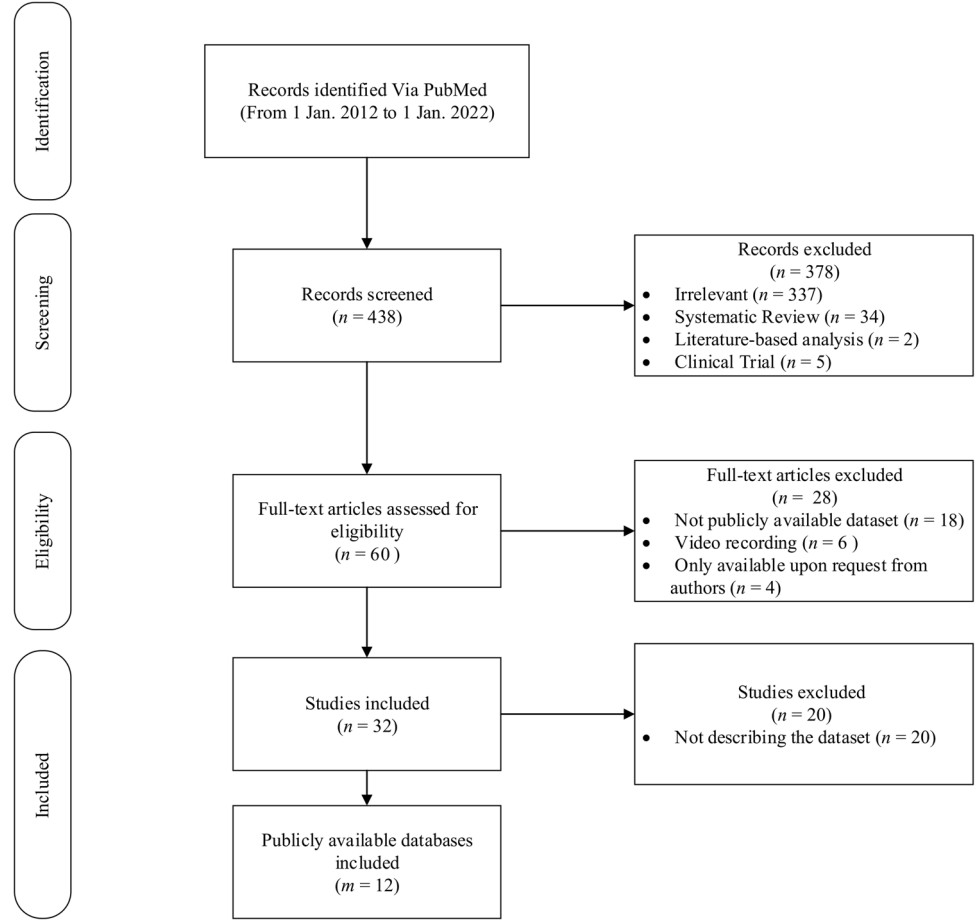

**Fig. 1 Flow chart of study identification, inclusion and exclusion criteria.** Presentation of the literature search and selection procedure together with the numbers of records at each stage sourced from PubMed from 1 January 2012 to 1 January 2022 using the following keywords: (("critical care"[MeSH Terms] OR ("critical"[All Fields] AND "care"[All Fields]) OR "critical care"[All Fields] OR ("oximetry"[MeSH Terms] OR "oximetry"[All Fields] OR ("oxygen"[All Fields] AND "saturation"[All Fields]) OR "oxygen saturation"[All Fields]) OR ("oximetry"[MeSH Terms] OR "oximetry"[All Fields] OR ("o2"[All Fields] AND "saturation"[All Fields]) OR "o2 saturation"[All Fields]) OR "PPG"[All Fields] OR ("photoplethysmogram"[All Fields] OR "photoplethysmograms"[All Fields]) OR ("photoplethysmography"[MeSH Terms] OR "photoplethysmography"[All Fields])) AND (("publicly"[All Fields] AND ("availabilities"[All Fields] OR "availability"[All Fields] OR "available"[All Fields])) OR ("freely"[All Fields] AND ("access"[All Fields] OR "accessed"[All Fields] OR "accesses"[All Fields] OR "accessibilities"[All Fields] OR "accessibility"[All Fields] OR "accessible"[All Fields] OR "accessing"[All Fields])))) AND ((humans[Filter]) AND (english[Filter])). Papers that met the inclusion criteria, discussing the development and publication of original pulse oximetry datasets, were selected for analysis. This resulted in 12 research papers representing 12 publicly available datasets. Here n refers to the number of studies, where m refers to the number of publicly available databases.

**Table 1 Summary of all the 12 publicly available datasets.**

| Title | Authors (year of publication) | Name of database | Number of subjects | Country of data location | Ethnicity is clearly stated or inferred | Ethnicity | Number of subjects based on ethnic group |
|---|---|---|---|---|---|---|---|
| A database to support development and evaluation of intelligent intensive care monitoring | Moody and Mark[12] (1996) | MIMIC I | 93 | US | Clearly stated | 2.4% Asian 9.1% Black 19.1% Other 70.3% White | 2 Asian 8 Black 18 Other 65 White |
| CapnoBase: signal database and tools to collect, share and annotate respiratory signals | Karlen et al.[15] (2010) | CapnoBase | 42 | Canada | Inferred | 11.0% Asian 3.5% Black 12.6% Other 72.9% White | 4 Asian 1 Black 6 Other 31 White |
| Multiparameter intelligent monitoring in intensive care II (MIMIC-II): a public-access intensive care unit database | Saeed et al.[10] (2011) | MIMIC-II | 32000 | US | Clearly stated | 2.5% Asian 10.6% Black 17.7% Other 69.2% White | 800 Asian 3392 Black 5664 Other 22144 White |
| University of Queensland vital signs dataset: development of an accessible repository of anesthesia patient monitoring data for research | Liu et al.[16] (2012) | University of Queensland Vital Signs | 32 | Australia | Inferred | 3.1% Asian 0.4% Black 27.7% Other 69.2% White | 1 Asian 0 Black 9 Other 22 White |
| TROIKA: a general framework for heart rate monitoring using wrist-type photoplethysmographic (PPG) signals during intensive physical exercise | Zhang et al.[11] (2015) | IEEEPPG | 12 | China | Inferred | 100.0% Asian 0.0% Black 0.0% Other 0.0% White | 12 Asian 0 Black 0 Other 0 White |
| MIMIC-III, a freely accessible critical care database | Johnson et al.[9] (2016) | MIMIC-III | 53423 | US | Clearly stated | 2.4% Asian 7.7% Black 18.6% Other 71.3% White | 1282 Asian 4113 Black 9937 Other 38091 White |
| An assessment of algorithms to estimate respiratory rate from the electrocardiogram and photoplethysmogram | Charlton et al.[14] (2016) | Vortal | 45 | UK | Inferred | 7.5% Asian 3.4% Black 9.1% Other 80.0% White | 4 Asian 2 Black 3 Other 36 White |
| Description of a database containing wrist PPG signals recorded during physical exercise with both accelerometer and gyroscope measures of motion | Jarchi and Casson[19] (2016) | Wrist PPG Signals Recorded during Exercise | 8 | UK | Inferred | 7.5% Asian 3.4% Black 0.1% Other 80.0% White | 1 Asian 1 Black 0 Other 6 White |
| Introducing WESAD, a multimodal dataset for wearable stress and affect detection | Schmidt et al.[13] (2018) | WESAD | 15 | Germany | Inferred | 2.5% Asian 1.0% Black 8.3% Other 88.2% White | 0 Asian 0 Black 2 Other 13 White |
| A new, short-recorded photoplethysmogram dataset for blood pressure monitoring in China | Liang et al.[18] (2018) | PPG-BP | 219 | China | Inferred | 92.9% Asian 0.0% Black 7.1% Other 0.0% White | 203 Asian 0 Black 16 Other 0 White |
| Deep PPG: large-scale heart rate estimation with convolutional neural networks | Reiss et al.[17] (2019) | PPG-DaLiA | 15 | Germany | Inferred | 2.5% Asian 1.0% Black 8.3% Other 88.2% White | 0 Asian 0 Black 2 Other 13 White |
| MIMIC-IV | Johnson et al.[20] (2021) | MIMIC-IV | 60000 | US | Clearly stated | 3.0% Asian 10.0% Black 10.0% Other 77.0% White | 1800 Asian 6000 Black 6000 Other 46200 White |

The numbers of subjects and distribution of different ethnic groups as identified or inferred in the publicly available datasets. In case of the absence of patient ethnicity information, the location of the authors' research institutions' orthe data collection location was used to infer ethnicity based on local statistics. For Vortal and Wrist PPG Signals Recorded during Exercise databases, we inferred the ethnicity of subjects based on the Institute of Race Relations (https://irr.org.uk/research/statistics/ethnicity-and-religion/). For CapnoBase database, we inferred the ethnicity of subjects based on Statistics Canada (https://www.statcan.gc.ca/en/start) reviewing "Census Profile, 2016 Census" (https://www12.statcan.gc.ca/census-recensement/2016/dp-pd/prof/details/page.cfm?Lang=E&Geo1=PR&Code1 = 01&Geo2=PR&Code2 = 01&Data=Count&SearchText=canada&SearchType=Begins&SearchPR=01&B1 = All&TABID = 1). As an example, the Black population accounts for 3.5% of Canada's total population (https://www.statcan.gc.ca/en/dai/smr08/2022/smr08_259). For University of Queensland Vital Signs, we inferred the ethnicity of subjects based on Australian Bureau of Statistics (https://www.abs.gov.au/). For WESAD and PPG-DaLiA databases, we inferred the ethnicity of subjects based on Statistisches Bundesamt (https://www.destatis.de/EN/Home/_node.html). For PPG-BP, we inferred the ethnicity of subjects based on National Bureau of Statistics of China (http://www.stats.gov.cn/english/).

We tested the statistical significance among all the subjects in the four databases considering a *p*-value <0.05 as statistically significant and analyzed the variance using a one-way ANOVA followed by post hoc test to provide simultaneous two-way interactions using the Tukey's honest significant difference criterion. The results indicated that there was a significant difference between the mean distributions of all racial groups; Asian and Black ($p = 0.021$), Asian and white ($p = 4.10 \times 10^{-14}$), and Black and white ($p = 5.01 \times 10^{-13}$). The same trend was observed between Other and Asian ($p = 9.43 \times 10^{-05}$), Other and Black

($p = 0.026$), and Other and white ($p = 4.82 \times 10^{-12}$). The results also suggested a higher proportion of white subjects compared to Asian, Black and other populations. These results demonstrate the existence of clear disparities in these key databases. Detailed results on the statistical separability tests for all pairs of demographic groups are provided in Table 2.

In the remaining databases in which ethnicity was not explicitly stated, the ethnic disparity is not known. However, if we examine the demographic statistics of each data set, based on location, we see that significant potential for disparity exists. For example, the Vortal dataset was collected in the UK in 2016, and the authors did not provide the race of each participant. Based on government records, we can infer the ethnic distributions based on UK ethnicity statistics: 7.5% Asian, 3.4% Black, 0.1% Other, and 80.0% white. The same method to infer ethnicity was used for the remaining databases, as shown in Table 1. Furthermore, since the racial groups were not clearly defined, it does suggest a lax

approach to the matter of constructing reference databases, mainly when applied to vascular optical measurement technology that can be influenced by skin color characteristics. White subjects appeared in all four MIMIC databases where the ethnicity was clearly stated, constituting an average of 73.19% of the total population. However, Black subjects only accounted for an average of 9.29% of the sample population. In addition, Asian subjects comprised an average of 2.67% of the total population investigated. Such distributions highlight the potential for racial and ethnic biases in algorithms and devices, leading to possible challenges in their wider application in medicine.

Our findings highlight clear disparities in pulse oximetry databases. As these biased databases would be used during the premarket phase to adjust pulse oximeter accuracy and to develop algorithms for oxygen saturation determination, they place subjects with darker skin pigmentation at increased risk of unrecognized health conditions[3]. Such health inequalities necessitate the development of new pulse oximeter databases with more racially balanced populations. Our recommendation does not deny the value of exploiting existing biased databases; rather, it attempts to benefit from using these publicly available databases when testing developed algorithms, as well as aiming for more balanced populations in future databases. Asian and Black populations have low representation in existing databases and it would also be beneficial to create an increased number of databases from different geographical regions.

Since last year, the US Food and Drug Administration has started to issue new guidelines to evaluate pre- and post-market pulse oximeters[3], and to increase awareness of racial and ethnic disparities that can affect the accuracy of pulse oximetry algorithms. As publicly-accessible databases are commonly used for the development of many biomedical algorithms and devices, our findings highlight the need to improve device algorithms and expand these databases to better represent a diversity of skin pigmentations regardless of the racial or ethnic group. Improving diversity in public databases would help improve the general accuracy of AI algorithms, especially for measurements that involve frequently life-threatening conditions such as COVID-19.

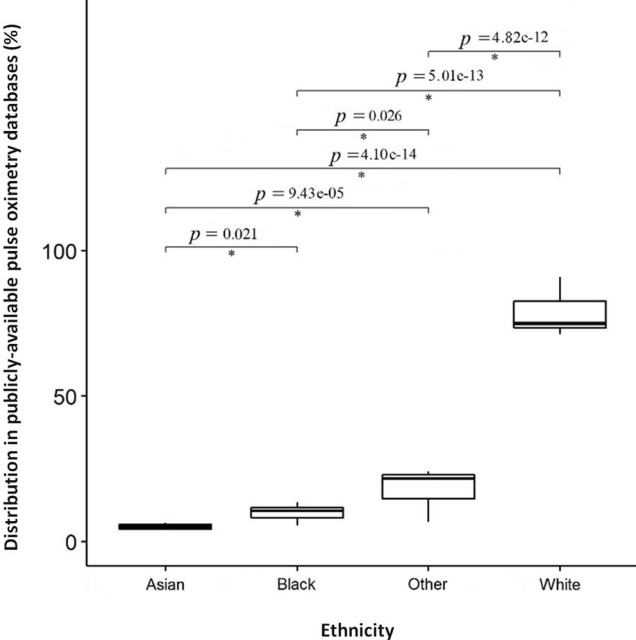

**Fig. 2 Box plots of the ethnic makeup by proportion in all databases.**
This figure combines all the databases used in the four publicly available pulse oximeter databases that clearly reported the distribution of ethnic groups. The data supports the hypothesis that disparities exist here. Significant differences are evident between white and Black ($p < 0.0001$), white and Asian ($p < 0.0001$), and Black and Asian populations ($p = 0.021$). All pairs of groups were tested by using a simultaneous pairwise Tukey test. The bottom and the top of the box are the 25th and 75th percentiles, and the line inside the box is the 50th percentile (median). Whiskers from minimum to maximum are determined with a 95% confidence interval.

## Data availability

Supplementary Data 1 contains source data for the main figures in this manuscript. Pulse oximetry databases can be accessed via the following links: MIMIC-I (https://www.physionet.org/content/mimicdb/1.0.0/); CapnoBase (https://dataverse.scholarsportal.info/dataverse/capnobase#:~:text=The%20CapnoBase%20benchmark%20dataset%20contains,that%20may%20arise%20during%20anesthesia.); MIMIC-II (https://archive.physionet.org/physiobank/database/mimic2wdb/); University of Queensland Vital Signs (https://outbox.eait.uq.edu.au/uqdliu3/uqvitalsignsdataset/index.html#:~:text=Introduction,at%20the%20Royal%20Adelaide%20Hospital.); IEEEPPG (https://zenodo.org/record/3902710#.YmsOVNrMKUk); MIMIC-III (https://physionet.org/content/mimiciii/1.4/); Vortal (https://peterhcharlton.github.io/RRest/vortal_dataset.html); Wrist PPG Signals Recorded during Exercise (https://physionet.org/content/wrist/1.0.0/); WESAD (https://archive.ics.uci.edu/ml/datasets/WESAD+%28Wearable+Stress+and+Affect+Detection%29); PPG-BP (https://figshare.com/articles/dataset/PPG-BP_Database_zip/5459299); PPG-DaLiA (https://archive.ics.uci.edu/ml/datasets/PPG-DaLiA); MIMIC-IV (https://physionet.org/content/mimiciv/1.0/).

**Table 2 Tukey simultaneous tests for differences of means.**

| Difference of levels | Difference of means | SE of difference | 95% CI | T-value | Adjusted *P*-value |
|---|---|---|---|---|---|
| Asian-black | 6.77 | 2.00 | (0.84, 12.71) | 3.39 | 0.02 |
| Asian-other | 13.78 | 2.00 | (7.84, 19.71) | 6.89 | <0.0001 |
| Asian-white | 69.38 | 2.00 | (63.44, 75.31) | 34.69 | <0.0001 |
| Black-other | 7.00 | 2.00 | (1.06, 12.94) | 3.50 | 0.02 |
| Black-white | 62.60 | 2.00 | (56.66, 68.54) | 31.30 | <0.0001 |
| Other-white | −55.60 | 2.00 | (−61.54, −49.66) | −27.80 | <0.0001 |

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

## Author contributions

M.E. designed and led this investigation. F.S., R.W., D.A., J.A., R.F., C.M. and M.E. conceived the study. All authors approved the final manuscript.

## Competing interests

The authors declare no competing interests.
