## [Peer Review File · Communications Medicine]

Reviewers' comments:

Reviewer #1 (Remarks to the Author):

The authors present analysis of the racial make-up of available databases containing information about pulse oximetry that could be used to create biomedical algorithms. The diversity of available databases is of general interest and their result, that such databases contain lower diversity is not very surprising yet still important.

It wasn't very clear how the authors performed the "comprehensive assessment" to find all the databases. How did the authors perform the search? What criteria were used to determine if a database should be included in this analysis? These details are important for determining the comprehensiveness of this search, and the validity of the conclusions.

Reviewer #2 (Remarks to the Author):

This study was prompted by a recent report which revealed a racial disparity in the accuracy of pulse oximetry readings. It reviews the results of twenty seven studies of pulse oximetry signals in subjects with hypertension, drawn from eight public databases and describes the discrepancies between the numbers of subjects of different skin colour. It also mentions that, of the total number of subjects in all eight databases, a large majority is light skinned. This is important and the paper makes this very clear, because the amplitude of photoplethysmographic signals is dependent on the degree of skin pigmentation. It follows that algorithms used in commercial systems calibrated for light skinned Caucasians, say, may not yield accurate SpO₂ readings for subjects with darker skin. Thus the conclusion of the paper, that consideration be given to the distribution of skin-colour of database subjects, is justified.

The main result of this report is that there are significant differences between the 8 databases in the ethnic (and thus skin colour) make-up of their subjects. This disparity would be less of a problem if the source data had been stratified according to ethnicity/skin colour. Presumably this was not the case but it would be useful if the question were briefly discussed. The authors recommend that future databases be ethnically balanced, but is this necessarily the only way to solve the problem? It might be better, when using the databases for algorithm calibration, for instance, to exploit the existing studies with large numbers of a particular skin colour as well as aiming for more balanced populations in future databases.

I have some questions about the statistical methods used. Firstly, a few words would be useful to explain why both parametric and non-parametric tests were used to seek differences in the composition of the databases. I note that the significance of both (at the 5% level) coincides in all but one case (European v East Asian). Which value should be accepted and why? Secondly, I am concerned about the dangers of multiple comparisons between pairs of groups, because this can give rise to contradictions, or at least, inconsistencies. Would 1-way ANOVA followed by a post-hoc test be a more appropriate way test the hypothesis? Finally, I am not clear why the percentages for each database listed in appendix C don't add up to 100.

Minor points.

- Second word of the text "oximeter"  "oximetry"

- As an adjective, should black be capitalised? In the main text it has been uniformly capitalised but is inconsistent in appendix A.
- In the first line of the penultimate paragraph, what does “warm color spectrum” refer to?

Ethnic Disparities in Pulse Oximetry Databases

Reviewer 1:

1. It wasn't very clear how the authors performed the "comprehensive assessment" to find all the databases. How did the authors perform the search? What criteria were used to determine if a database should be included in this analysis?

Author reply: Thank you for your valuable question. A comprehensive assessment was implemented by searching for all publicly available databases published in the literature over the last decade.

Author action: We have added a flow chart of study identification, inclusion and exclusion criteria to show a comprehensive search for publicly available PPG datasets as Figure 1 and have improved the wording, accordingly, starting at *line 50*, as follows:

“To explore how racial diversity impacts common databases used for academic medical research, we conducted a comprehensive assessment of accessible databases from 1 January 2010 to 1 January 2021 using PubMed consisting of Medical Subject Headings (MeSH) terms and Title/Abstract keywords. Applying the inclusion and exclusion criteria as defined in Figure 1 resulted in 11 research articles describing 11 publicly available datasets to assess different medical conditions using pulse oximeter data.”

Reviewer 2:

1. The authors recommend that future databases be ethnically balanced, but is this necessarily the only way to solve the problem? It might be better, when using the databases for algorithm calibration, for instance, to exploit the existing studies with large numbers of a particular skin colour as well as aiming for more balanced populations in future databases.

Author reply: Thank you for your valuable feedback. Using existing databases while developing more diversified ones in future has also been the intention of our recommendation.

Author action: We added the following text to clarify this point, starting at *line 127*, as follows:

“Our recommendation does not deny the value of exploiting existing biased databases; rather, it attempts to benefit from using these publicly available databases when testing developed algorithm, as well as aiming for more balanced populations in future databases creation”

2. Why both parametric and non-parametric tests were used to seek differences in the composition of the databases. I note that the significance of both (at the 5% level) coincides in all but one case (European v East Asian). Which value should be accepted and why?

Author reply: Thank you for your valuable feedback.

Author action: We removed both parametric and non-parametric tests and then added the 1-way ANOVA followed by a post-hoc test, as per reviewer suggestion in the next point.

3. I am concerned about the dangers of multiple comparisons between pairs of groups, because this can give rise to contradictions, or at least, inconsistencies. Would 1-way ANOVA followed by a post-hoc test be a more appropriate way test the hypothesis?

Author reply: Thank you for your valuable feedback. We conducted the 1-way ANOVA followed by a post-hoc.

Author action 1: We added the following paragraph to clarify this point, starting at line 103, as follows:

“We tested the statistical significance among all the subjects in the databases using a significance level of 0.05 and analyzed the variance using a one-way ANOVA followed by post-hoc test to provide simultaneous two-way interactions using the Tukey's honest significant difference criterion. The results indicated that there was a significant difference between the mean distributions of those with European ancestry and African American ancestry ($p = 2.12 \times 10^{-05}$). The same trend was observed for those with European and South Asian ancestry ($p = 2.24 \times 10^{-05}$) and European and East Asian ancestry ($p = 3.22 \times 10^{-04}$). These results suggested a higher proportion of those with European ancestry (White population) compared to those with African American (Black population), South Asian, and East Asian ancestry. These results demonstrate the existence of clear racial disparities in the databases. Detailed results on the statistical separability tests for all pairs of racial groups are provided in Table 2.”

4. I am not clear why the percentages for each database listed in appendix C don't add up to 100.

Author reply: Thank you for your valuable feedback.

Author action: We removed appendix C and created table 1 which shows the percentages more clearly.

5. In the first line of the penultimate paragraph, what does “warm color spectrum” refer to?

Author reply: Thank you for your valuable feedback and bringing this to our attention.

Author action: For readability, we have now removed this wording.

Reviewers' comments:

Reviewer #1 (Remarks to the Author):

The authors have addressed my concerns with the manuscript

Reviewer #2 (Remarks to the Author):

Some of my questions have been satisfactorily answered. However in a couple of cases a change has been made without an answer being provided.

These are:

Point 4. Why did the percentages in appendix C not add up to 100? You have added table 1 in which the figures I have checked do add up to 100, but you have not answered the question.

Point 5. You have removed the phrase "warm colour spectrum" but presumably it meant something. What did it mean?

There are a couple of minor points:

No explicit reply to the minor points about capitalising the word "black", also "white". Why should it be capitalised? I have no objection in principle but I would like to know the reason. Presumably, you wish to treat both words as proper nouns as you have with South Asian etc., but for consistency it might be better to use the descriptions Afro-Caribbean (or African American) and European instead. This is, of course a very small point and you may consider my suggestion to be overly pedantic. If so, OK.

Legend to table 1. Is there a missing "was" between "groups" and "identified", (line 79)?

Line 123. "criteria"  "criterion"

Line 129. A missing "a" between "testing" and "developed"?

Line 144. Add "the" between "final" and "manuscript".

Ethnic Disparities in Pulse Oximetry Databases

Reviewer 2 rebuttal:

1. Why did the percentages in appendix C not add up to 100? You have added table 1 in which the figures I have checked do add up to 100, but you have not answered the question.

Author reply: Thank you for your valuable question. May I clarify please. The remaining percentage for the racial databases in that table was associated with the “Other” category. Previously, the races under this category were excluded from our analysis.

Author action: We removed appendix C and created table 1 which now shows the percentages more clearly.

2. You have removed the phrase “warm colour spectrum” but presumably it meant something. What did it mean?

Author reply: Thank you for your valuable question about this point. Previously the “warm color spectrum” was referring to a heat map that was removed from a previously submitted version of the manuscript.

Author action: We have removed the heat map and the queried wording linked to this.

3. No explicit reply to the minor points about capitalising the word “black”, also “white”. Why should it be capitalised? I have no objection in principle but I would like to know the reason. Presumably, you wish to treat both words as proper nouns as you have with South Asian etc., but for consistency it might be better to use the descriptions Afro-Caribbean (or African American) and European instead. This is, of course a very small point and you may consider my suggestion to be overly pedantic. If so, OK.

Author reply: Thank you for this point also. It is very important of course to get this right. As recently requested by Associate Editor Ben Abbott, we have changed our main racial groups analysis to be based on only those existing patients’ race records which are clearly reported in the publicly available databases (Asian, Black, White, Other). We have highlighted in the text that for the remaining databases there is a lack of race information and explained how this might impede studies using these databases.

Author action: Based on the following comment from reviewer 1, we updated the manuscript. “According to our Nature house style, ‘Black’ should be capitalized and ‘white’ should not.”

REVIEWERS' COMMENTS:

Reviewer #1 (Remarks to the Author):

The authors have addressed all concerns at this point.

Reviewer #2 (Remarks to the Author):

My earlier questions are answered. Thank you.

I think the newly added table 1 makes the limitations relating to the inferred skin colour adequately clear and the decision to omit these studies from the statistical analysis is acceptable.

The newly added sentence near the bottom of page 1 doesn't quite make sense.

To prevent potential disparities in the calibration and accuracy of such medical instruments (i.e. pulse oximetry devices) and their algorithms, the pulse oximeter signals are representative of the diverse populations on which these devices are used.

Is a phrase missing after the comma? Perhaps something like "it is important that"?

REVIEWERS' COMMENTS:

Reviewer #1 (Remarks to the Author):

The authors have addressed all concerns at this point.

Authors' response: We thank the reviewer for their time and for the positive feedback.

Authors' action: None

Reviewer #2 (Remarks to the Author):

My earlier questions are answered. Thank you.

I think the newly added table 1 makes the limitations relating to the inferred skin colour adequately clear and the decision to omit these studies from the statistical analysis is acceptable.

Authors' response: We thank the reviewer for their time and for the positive feedback.

Authors' action: None

The newly added sentence near the bottom of page 1 doesn't quite make sense.

To prevent potential disparities in the calibration and accuracy of such medical instruments (i.e. pulse oximetry devices) and their algorithms, the pulse oximeter signals are representative of the diverse populations on which these devices are used.

Is a phrase missing after the comma? Perhaps something like "it is important that"?

Authors' response: We thank the reviewer for the valuable suggestion.

Authors' action: We modified the sentence to read as follows.

"To prevent potential disparities in the calibration and accuracy of pulse oximetry devices and their algorithms, the pulse oximeter signals within such public databases need to be representative of the diverse populations on which these devices are used."